# Adaptive Batch Size for Safe Policy Gradients

**Matteo Papini**
DEIB
Politecnico di Milano, Italy
matteo.papini@polimi.it

**Matteo Pirotta**
SequeL Team
Inria Lille, France
matteo.pirotta@inria.fr

**Marcello Restelli**
DEIB
Politecnico di Milano, Italy
marcello.restelli@polimi.it

## Abstract

Policy gradient methods are among the best Reinforcement Learning (RL) techniques to solve complex control problems. In real-world RL applications, it is common to have a good initial policy whose performance needs to be improved and it may not be acceptable to try bad policies during the learning process. Although several methods for choosing the step size exist, research paid less attention to determine the batch size, that is the number of samples used to estimate the gradient direction for each update of the policy parameters. In this paper, we propose a set of methods to jointly optimize the step and the batch sizes that guarantee (with high probability) to improve the policy performance after each update. Besides providing theoretical guarantees, we show numerical simulations to analyse the behaviour of our methods.

## 1 Introduction

In many real-world sequential decision-making problems (e.g., industrial robotics, natural resource management, smart grids), engineers have developed automatic control policies usually derived from modelling approaches. The performance of such policies strictly depends on the model accuracy that for some tasks (e.g., financial applications) may be quite poor. Furthermore, even when accurate models are available and good control policies are obtained, their performance may degrade over time due to the non-stationary dynamics of the problem, thus requiring human intervention to adjust the policy parameters (think about equipment wear in smart manufacturing). In such scenarios, Reinforcement Learning (RL) techniques represent an interesting solution to get an online optimization of the control policies and to hinder the performance loss caused by unpredictable environment changes, thus allowing to improve the autonomy of the control system.

In the last years, several RL studies [1, 2, 3, 4, 5, 6, 7] have shown that policy-search methods can effectively be employed to solve complex control tasks (e.g., robotic ones) due to their capabilities to handle high-dimensional continuous problems, face uncertain and partial observations of the state, and incorporate prior knowledge about the problem by means of the definition of a proper policy model whose parameters need to be optimized (refer to [8, 9] for recent surveys). This last property is particularly appealing when the reinforcement learning algorithm needs to operate online in scenarios where bad exploratory policies may damage the system. A proper design of the policy model may allow excluding such policies. On the other hand, in order to speed up the learning process, most RL methods need to explore the policy space by executing policies that may be worse than the initial one. This is not acceptable in many relevant applications. Under this perspective, we are interested in developing RL methods that are (in high probability) monotonically improving.

Inspired by the *conservative policy iteration* approach [10], recently, new advances have been done in the field of approximate policy iteration algorithms [11, 12], obtaining methods that can learn faster while still giving statistical guarantees of improvement after each policy update [13, 14]. These methods are usually referred to as conservative, monotonically improving, or safe (as we do in this paper). These ideas have been exploited also for deriving novel safe policy-search approaches [15, 16,

17, 18, 19] that have obtained significant empirical results. In particular, policy-gradient methods are among the most commonly used RL techniques to solve complex high-dimensional tasks. Up to now, works on safe policy gradients [15, 16] have focused mainly on the choice of the step size, a parameter that significantly affects the speed and stability of gradient methods. By adopting small enough step sizes, one can limit oscillations and avoid worsening updates, but the consequent reduction of the learning rate is paid on the long term as a poor overall performance. On the other hand, as we will show in this paper, there is another parameter that plays an important role in the definition of safe policy gradient approaches: the batch size (i.e., the number of samples used to estimate the gradient). So far, the optimization of the batch size has not been considered in the RL literature. The batch size, besides conditioning the optimal step size, has a non-negligible impact on the speed of improvement when samples are trajectories performed on the actual system. In the present paper, we inquire the relationship between the step size and the batch size, showing an interesting duality. Focusing on Gaussian policies, we make a first attempt at developing practical methods aimed at achieving the best average performance in the long term, by jointly optimizing both meta-parameters. After providing some background in Section 2, in Section 3 we improve an existing adaptive step-size method [15]. Building on this result, in Section 4 we derive the main result on the batch size, proposing jointly adaptive methods. Finally, in Section 5 we empirically analyse the behaviour of the proposed methods on a simple simulated control task.

## 2 Preliminaries

A discrete-time continuous Markov decision process (MDP) is a tuple $\langle \mathcal{S}, \mathcal{A}, \mathcal{P}, \mathcal{R}, \gamma, \mu \rangle$, where $\mathcal{S}$ is the continuous state space, $\mathcal{A}$ is the continuous action space, $\mathcal{P}$ is a Markovian transition model where $\mathcal{P}(s'|s,a)$ defines the transition density between states $s$ and $s'$ under action $a$, $\mathcal{R} : \mathcal{S} \times \mathcal{A} \rightarrow [-R, R]$ is the reward function, such that $\mathcal{R}(s,a)$ is the expected immediate reward for the state-action pair $(s, a)$ and $R$ is the maximum absolute reward value, $\gamma \in [0, 1)$ is the discount factor for future rewards and $\mu$ is the initial state distribution. A policy is defined as a density distribution $\pi(\cdot|s)$ that, for each state $s$, specifies the density distribution over action space $\mathcal{A}$. We consider infinite horizon problems where the future rewards are exponentially discounted with $\gamma$. For each state-action pair $(s, a)$, the utility of taking action $a$ in state $s$ and then following a stationary policy $\pi$ is defined as:

$$Q^\pi(s, a) = \mathcal{R}(s, a) + \gamma \int_{\mathcal{S}} \mathcal{P}(s'|s, a) \int_{\mathcal{A}} \pi(a'|s') Q^\pi(s', a') \mathrm{d}a' \mathrm{d}s'.$$

Policies can be ranked by their expected discounted reward starting from initial state distribution $\mu$:

$$J_\mu^\pi = \int_{\mathcal{S}} \mu(s) \int_{\mathcal{A}} \pi(a \mid s) Q^\pi(s, a) \mathrm{d}a \mathrm{d}s = \int_{\mathcal{S}} d_\mu^\pi(s) \int_{\mathcal{A}} \pi(a|s) \mathcal{R}(s, a) \mathrm{d}a \mathrm{d}s,$$

where $d_\mu^\pi(s) = (1 - \gamma) \sum_{t=0}^{\infty} \gamma^t Pr(s_t = s|\pi, \mu)$ is the $\gamma$-discounted future state distribution for a starting state distribution $\mu$ [2]. In the following, we will often refer to $J_\mu^\pi$ as the performance of policy $\pi$. Solving an MDP means finding a policy $\pi^*$ maximizing $J_\mu^\pi$. We consider the problem of finding a policy that maximizes the expected discounted reward over a class of parametrized policies $\Pi_{\boldsymbol{\theta}} = \{\pi_{\boldsymbol{\theta}} : \boldsymbol{\theta} \in \mathbb{R}^m\}$. A particular class of parametrized policies is the Gaussian policy model with standard deviation $\sigma$ and mean linear in the state features $\phi(\cdot)$:

$$\pi(a|s, \boldsymbol{\theta}) = \frac{1}{\sqrt{2\pi\sigma^2}} \exp\left(-\frac{1}{2}\left(\frac{a - \boldsymbol{\theta}^T \phi(s)}{\sigma}\right)^2\right),$$

which is a common choice for MDPs with continuous actions. The exact gradient of the expected discounted reward w.r.t. the policy parameters [2] is:

$$\nabla_{\boldsymbol{\theta}} J_\mu(\boldsymbol{\theta}) = \frac{1}{1 - \gamma} \int_{\mathcal{S}} d_\mu^{\pi_{\boldsymbol{\theta}}}(s) \int_{\mathcal{A}} \nabla_{\boldsymbol{\theta}} \pi(a|s, \boldsymbol{\theta}) Q^{\pi_{\boldsymbol{\theta}}}(s, a) \mathrm{d}a \mathrm{d}s.$$

In most commonly used policy gradient methods, the policy parameters are updated by following the direction of the gradient of the expected discounted reward: $\boldsymbol{\theta}' = \boldsymbol{\theta} + \alpha \nabla_{\boldsymbol{\theta}} J_\mu(\boldsymbol{\theta})$, where $\alpha \geq 0$ is a scalar step size. In the following we will denote with $\|\nabla_{\boldsymbol{\theta}} J_\mu(\boldsymbol{\theta})\|_p$ the $L^p$-norm of the policy gradient.

# 3 Non-Scalar Step Size for Gaussian Policies

Before starting to optimize the batch size for the gradient estimation, in this section we extend the results in [15] to the case of a non-scalar step size, showing that, focusing on the Gaussian policy model, such extension guarantees a larger performance improvement than the one obtained in [15]. Furthermore, this result significantly simplifies the closed-form solutions obtained for the optimization of the batch size described in the following sections. In Section 3.1 we stick to the theoretical setting in which the gradient is known exactly, while in Section 3.2 we take into account the estimation error.

## 3.1 Exact Framework

The idea is to have a separate adaptive step size $\alpha_i$ for each component $\theta_i$ of $\boldsymbol{\theta}$. For notational convenience, we define a non-scalar step size as a diagonal matrix $\Lambda = diag(\alpha_1, \alpha_2, \ldots, \alpha_m)$ with $\alpha_i \geq 0$ for $i = 1, \ldots, m$. The policy parameters can be updated as:

$$\boldsymbol{\theta}' = \boldsymbol{\theta} + \Lambda \nabla_{\boldsymbol{\theta}} J_\mu(\boldsymbol{\theta}).$$

Note that the direction of the update can differ from the gradient direction. Since the $\alpha_i$ are non-negative, the absolute angular difference is never more than $\pi/2$. The traditional scalar step-size update can be seen as a special case where $\Lambda = \alpha I$.

**Assumption 3.1.** *State features are uniformly bounded:* $|\phi_i(s)| \leq M_\phi, \forall s \in \mathcal{S}, \forall i = 1, \ldots, m.$

By adapting Theorem 4.3 in [15] to the new parameter update, we obtain a lower bound on the policy performance improvement:

**Lemma 3.2.** *For any initial state distribution $\mu$ and any pair of stationary Gaussian policies $\pi_{\boldsymbol{\theta}} \sim \mathcal{N}(\boldsymbol{\theta}^T \boldsymbol{\phi}(s), \sigma^2)$ and $\pi_{\boldsymbol{\theta}'} \sim \mathcal{N}(\boldsymbol{\theta}'^T \boldsymbol{\phi}(s), \sigma^2)$, so that $\boldsymbol{\theta}' = \boldsymbol{\theta} + \Lambda \nabla_{\boldsymbol{\theta}} J_\mu(\boldsymbol{\theta})$, and under Assumption 3.1, the difference between the performance of $\pi_{\boldsymbol{\theta}'}$ and the one of $\pi_{\boldsymbol{\theta}}$ can be bounded below as follows:*

$$J_\mu(\boldsymbol{\theta}') - J_\mu(\boldsymbol{\theta}) \geq \nabla_{\boldsymbol{\theta}} J_\mu(\boldsymbol{\theta})^T \Lambda \nabla_{\boldsymbol{\theta}} J_\mu(\boldsymbol{\theta}) - \frac{\|\Lambda \nabla_{\boldsymbol{\theta}} J_\mu(\boldsymbol{\theta})\|_1^2 M_\phi^2}{(1-\gamma)\sigma^2} \left( \frac{1}{\sqrt{2\pi}\sigma} \int_{\mathcal{S}} d_\mu^{\pi_{\boldsymbol{\theta}}}(s) \int_{\mathcal{A}} Q^{\pi_{\boldsymbol{\theta}}}(s,a) \mathrm{d}a \mathrm{d}s + \frac{\gamma \|Q^{\pi_{\boldsymbol{\theta}}}\|_\infty}{2(1-\gamma)} \right),$$

*where $\|Q^{\pi_{\boldsymbol{\theta}}}\|_\infty$ is the supremum norm of the Q-function:* $\|Q^{\pi_{\boldsymbol{\theta}}}\|_\infty = \sup_{s \in \mathcal{S}, a \in \mathcal{A}} Q^{\pi_{\boldsymbol{\theta}}}(s,a).$

The above bound requires us to compute the Q-function explicitly, but this is often not possible in real-world applications. We now consider a simplified (although less tight) version of the bound that does not have this requirement, which is an adaptation of Corollary 5.1 in [15]:

**Theorem 3.3.** *For any initial state distribution $\mu$ and any pair of stationary Gaussian policies $\pi_{\boldsymbol{\theta}} \sim \mathcal{N}(\boldsymbol{\theta}^T \boldsymbol{\phi}(s), \sigma^2)$ and $\pi_{\boldsymbol{\theta}'} \sim \mathcal{N}(\boldsymbol{\theta}'^T \boldsymbol{\phi}(s), \sigma^2)$, so that $\boldsymbol{\theta}' = \boldsymbol{\theta} + \Lambda \nabla_{\boldsymbol{\theta}} J_\mu(\boldsymbol{\theta})$, and under Assumption 3.1, the difference between the performance of $\pi_{\boldsymbol{\theta}'}$ and the one of $\pi_{\boldsymbol{\theta}}$ can be bounded below as follows:*

$$J_\mu(\boldsymbol{\theta}') - J_\mu(\boldsymbol{\theta}) \geq \nabla_{\boldsymbol{\theta}} J_\mu(\boldsymbol{\theta})^T \Lambda \nabla_{\boldsymbol{\theta}} J_\mu(\boldsymbol{\theta}) - c \|\Lambda \nabla_{\boldsymbol{\theta}} J_\mu(\boldsymbol{\theta})\|_1^2,$$

*where $c = \frac{R M_\phi^2}{(1-\gamma)^2 \sigma^2} \left( \frac{|\mathcal{A}|}{\sqrt{2\pi}\sigma} + \frac{\gamma}{2(1-\gamma)} \right)$ and $|\mathcal{A}|$ is the volume of the action space.*

We then find the step size $\Lambda^*$ that maximizes this lower bound under the natural constraint $\alpha_i \geq 0 \, \forall i = 1, \ldots, m$. The derivation is not trivial and is provided in Appendix A.

**Corollary 3.4.** *The lower bound of Theorem 3.3 is maximized by the following non-scalar step size:*

$$\alpha_k^* = \begin{cases} \frac{1}{2c} & \text{if } k = \min\left\{ \arg\max_i |\nabla_{\theta_i} J_\mu(\boldsymbol{\theta})| \right\}, \\ 0 & \text{otherwise}, \end{cases}$$

*which guarantees the following performance improvement:* $J_\mu(\boldsymbol{\theta}') - J_\mu(\boldsymbol{\theta}) \geq \frac{\|\nabla_{\boldsymbol{\theta}} J_\mu(\boldsymbol{\theta})\|_\infty^2}{4c}.$

Note that update induced by the obtained $\Lambda^*$ corresponds to employing a constant, scalar step size to update just the parameter corresponding to the largest absolute gradient component. This method is known in the literature as *greedy coordinate descent*. Convergence of this algorithm to a local

optimum is guaranteed for small step sizes, as shown in [20]. Note also that the way in which the index is selected in case of multiple maxima (here min) is arbitrary, see the proof of Corollary 3.4 for details. We now propose an intuitive explanation of our result: the employed performance lower bound ultimately derives from Corollary 3.6 in [13]. From the original bound, one can easily see that the positive part accounts to the average advantage of the new policy over the old one, while the negative part penalizes large parameter updates, which may result in overshooting. Updating just the parameter corresponding to the larger policy gradient component represents an intuitive trade-off between these two objectives. We now show that this result represents an improvement w.r.t. the adaptive scalar step size proposed in [15] for the current setting:

**Corollary 3.5.** *Under identical hypotheses, the performance improvement guaranteed by Corollary 3.4 is never less than the one guaranteed by Corollary 5.1 in [15], i.e.:*

$$\frac{\|\nabla_{\boldsymbol{\theta}} J_{\mu}(\boldsymbol{\theta})\|_{\infty}^2}{4c} \geq \frac{\|\nabla_{\boldsymbol{\theta}} J_{\mu}(\boldsymbol{\theta})\|_2^4}{4c \|\nabla_{\boldsymbol{\theta}} J_{\mu}(\boldsymbol{\theta})\|_1^2}.$$

This corollary derives from the trivial norm inequality $\|\nabla_{\boldsymbol{\theta}} J_{\mu}(\boldsymbol{\theta})\|_{\infty} \|\nabla_{\boldsymbol{\theta}} J_{\mu}(\boldsymbol{\theta})\|_1 \geq \|\nabla_{\boldsymbol{\theta}} J_{\mu}(\boldsymbol{\theta})\|_2^2$.

## 3.2 Approximate Framework

We now turn to the more realistic case in which the policy gradient, $\nabla_{\boldsymbol{\theta}} J_{\mu}(\boldsymbol{\theta})$, is not known, and has to be estimated from a finite number of trajectory samples. A performance improvement can still be guaranteed with high probability. To adapt the result of Theorem 3.3 to the stochastic gradient case, we need both a lower bound on the policy gradient estimate $\hat{\nabla}_{\boldsymbol{\theta}} J_{\mu}(\boldsymbol{\theta})$:

$$\underline{\hat{\nabla}_{\boldsymbol{\theta}} J_{\mu}}(\boldsymbol{\theta}) = \max(|\hat{\nabla}_{\boldsymbol{\theta}} J_{\mu}(\boldsymbol{\theta})| - \boldsymbol{\epsilon}, \mathbf{0})$$

(where the maximum is component-wise) and an upper bound:

$$\overline{\hat{\nabla}_{\boldsymbol{\theta}} J_{\mu}}(\boldsymbol{\theta}) = |\hat{\nabla}_{\boldsymbol{\theta}} J_{\mu}(\boldsymbol{\theta})| + \boldsymbol{\epsilon}$$

where $\boldsymbol{\epsilon} = [\epsilon_1, \dots, \epsilon_m]$, and $\epsilon_i$ is an upper bound on the approximation error of $\nabla_{\theta_i} J_{\mu}(\boldsymbol{\theta})$ with probability at least $1 - \delta$. We can now state the following:

**Theorem 3.6.** *Under the same assumptions of Theorem 3.3, and provided that a policy gradient estimate $\hat{\nabla}_{\boldsymbol{\theta}} J_{\mu}(\boldsymbol{\theta})$ is available, so that $\mathbb{P}(|\nabla_{\theta_i} J_{\mu}(\boldsymbol{\theta}) - \hat{\nabla}_{\theta_i} J_{\mu}(\boldsymbol{\theta})| \geq \epsilon_i) \leq \delta$, $\forall i = 1, \dots, m$, the difference between the performance of $\pi_{\boldsymbol{\theta}'}$ and the one of $\pi_{\boldsymbol{\theta}}$ can be bounded below with probability at least $(1 - \delta)^m$ as follows:*

$$J_{\mu}(\boldsymbol{\theta}') - J_{\mu}(\boldsymbol{\theta}) \geq \underline{\hat{\nabla}_{\boldsymbol{\theta}} J_{\mu}}(\boldsymbol{\theta})^T \Lambda \underline{\hat{\nabla}_{\boldsymbol{\theta}} J_{\mu}}(\boldsymbol{\theta}) - c \left\| \Lambda \overline{\hat{\nabla}_{\boldsymbol{\theta}} J_{\mu}}(\boldsymbol{\theta}) \right\|_1^2,$$

*where $c$ is defined as in Theorem 3.3.*

To derive the optimal step size, we first restrict our analysis to the case in which $\epsilon_1 = \epsilon_2 = \dots = \epsilon_m \triangleq \epsilon$. We call this common estimation error $\epsilon$. This comes naturally in the following section, where we use concentration bounds to give an expression for $\epsilon$. However, it is always possible to define a common error by $\epsilon = \max_i \epsilon_i$. We then need the following assumption:

**Assumption 3.7.** *At least one component of the policy gradient estimate is, in absolute value, no less than the approximation error:* $\left\| \hat{\nabla}_{\boldsymbol{\theta}} J_{\mu}(\boldsymbol{\theta}) \right\|_{\infty} \geq \epsilon$.

The violation of the above assumption can be used as a stopping condition since it prevents to guarantee any performance improvement. We can now state the following (the derivation is similar to the one of Corollary 3.5 and is, again, left to Appendix A):

**Corollary 3.8.** *The performance lower bound of Theorem 3.6 is maximized under Assumption 3.7 by the following non-scalar step size:*

$$\alpha_k^* = \begin{cases} \frac{\left(\left\|\hat{\nabla}_{\boldsymbol{\theta}} J_{\mu}(\boldsymbol{\theta})\right\|_{\infty} - \epsilon\right)^2}{2c\left(\left\|\hat{\nabla}_{\boldsymbol{\theta}} J_{\mu}(\boldsymbol{\theta})\right\|_{\infty} + \epsilon\right)^2} & \text{if } k = \min\left\{\arg\max_i |\hat{\nabla}_{\theta_i} J_{\mu}(\boldsymbol{\theta})|\right\}, \\ 0 & \text{otherwise,} \end{cases}$$

*which guarantees with probability $(1 - \delta)^m$ a performance improvement*

$$J_\mu(\boldsymbol{\theta}') - J_\mu(\boldsymbol{\theta}) \geq \frac{\left(\left\|\hat{\nabla}_{\boldsymbol{\theta}} J_\mu(\boldsymbol{\theta})\right\|_\infty - \epsilon\right)^4}{4c\left(\left\|\hat{\nabla}_{\boldsymbol{\theta}} J_\mu(\boldsymbol{\theta})\right\|_\infty + \epsilon\right)^2}.$$

## 4  Adaptive Batch Size

In this section we jointly optimize the step size for parameter updates and the batch size for policy gradient estimation, taking into consideration the cost of collecting sample trajectories. We call $N$ the batch size, i.e., the number of trajectories sampled to compute the policy gradient estimate $\hat{\nabla}_{\boldsymbol{\theta}} J_\mu(\boldsymbol{\theta})$ at each parameter update. We define the following cost-sensitive performance improvement measure:

**Definition 4.1.** Cost-sensitive performance improvement measure $\Upsilon_\delta$ is defined as:

$$\Upsilon_\delta(\Lambda, N) := \frac{B_\delta(\Lambda, N)}{N},$$

where $B_\delta$ is the high probability lower bound on performance improvement given in Theorem 3.6.

The rationale behind this choice of performance measure is to maximize the performance improvement *per sample trajectory*. Using larger batch sizes leads to more accurate policy updates, but the gained performance improvement is spread over a larger number of trials. This is particularly relevant in real-world online applications, where the collection of more samples with a sub-optimal policy affects the overall performance and must be justified by a greater improvement in the learned policy. By defining $\Upsilon_\delta$ in this way, we can control the improvement provided, on average, by each collected sample. We now show how to jointly select the step size $\Lambda$ and the batch size $N$ so as to maximize $\Upsilon_\delta$. Notice that the dependence of $B_\delta$ on $N$ is entirely through $\epsilon$, whose expression depends on which concentration bound is considered. We first restrict our analysis to concentration bounds that allow to express $\epsilon$ as follows:

**Assumption 4.1.** *The per-component policy gradient estimation error made by averaging over $N$ sample trajectories can be bounded with probability at least $1 - \delta$ by:*

$$\epsilon(N) = \frac{d_\delta}{\sqrt{N}},$$

*where $d_\delta$ is a constant w.r.t. $N$.*

This class of inequalities includes well-known concentration bounds such as Chebyshev's and Hoeffding's. Under Assumption 4.1 $\Upsilon_\delta$ can be optimized in closed form:

**Theorem 4.2.** *Under the hypotheses of Theorem 3.3 and Assumption 4.1, the cost-sensitive performance improvement measure $\Upsilon_\delta$, as defined in Definition 4.1, is maximized by the following step size and batch size:*

$$\alpha_k^* = \begin{cases} \frac{(13 - 3\sqrt{17})}{4c} & \text{if } k = \min\left\{\arg\max_i |\hat{\nabla}_{\theta_i} J_\mu(\boldsymbol{\theta})|\right\}, \\ 0 & \text{otherwise}, \end{cases} \qquad N^* = \left\lceil \frac{(13 + 3\sqrt{17})d_\delta^2}{2\left\|\hat{\nabla}_{\boldsymbol{\theta}} J_\mu(\boldsymbol{\theta})\right\|_\infty^2} \right\rceil,$$

*where $c = \frac{RM_\phi^2}{(1-\gamma)^2\sigma^2}\left(\frac{|\mathcal{A}|}{\sqrt{2\pi}\sigma} + \frac{\gamma}{2(1-\gamma)}\right)$. This choice guarantees with probability $(1 - \delta)^m$ a performance improvement of:*

$$J_\mu(\boldsymbol{\theta}') - J_\mu(\boldsymbol{\theta}) \geq \frac{393 - 95\sqrt{17}}{8}\left\|\hat{\nabla}_{\boldsymbol{\theta}} J_\mu(\boldsymbol{\theta})\right\|_\infty^2 \geq 0.16\left\|\hat{\nabla}_{\boldsymbol{\theta}} J_\mu(\boldsymbol{\theta})\right\|_\infty^2.$$

Notice that, under Assumption 4.1, Assumption 3.7 can be restated as $N \geq \frac{d_\delta^2}{\left\|\hat{\nabla}_{\boldsymbol{\theta}} J_\mu(\boldsymbol{\theta})\right\|_\infty^2}$, which is always verified by the proposed $N^*$. This means that the adaptive batch size never allows an estimation error larger than the gradient estimate. Another peculiarity of this result is that the step size is constant, in the sense that its value does not depend on the gradient estimate. This can be

explained in terms of a duality between step size and batch size: in other conservative adaptive-step size approaches, such as the one proposed with Theorem 4.2, the step size is kept small to counteract policy updates that are too off due to bad gradient estimates. When also the batch size is made adaptive, a sufficient number of sample trajectories can be taken to keep the policy update on track even with a constant-valued step size. Note that, in this formulation, the batch size selection process is always one step behind the gradient estimation. A possible synchronous implementation is to update $N^*$ each time a trajectory is performed, using all the data collected since the last learning step. As soon as the number of trajectories performed in the current learning iteration is larger than or equal to $N^*$, a new learning step is performed.

We now consider some concentration bounds in more detail: we provide the values for $d_\delta$, while the full expressions for $N^*$ can be found in Appendix B.

## 4.1 Chebyshev's Bound

By using the sample mean version of Chebyshev's bound we obtain:

$$d_\delta = \sqrt{\frac{Var[\tilde{\nabla}_{\theta_i} J_\mu(\boldsymbol{\theta})]}{\delta}},$$

where $\tilde{\nabla}_{\theta_i} J_\mu(\boldsymbol{\theta})$ is the policy gradient approximator (from a single sample trajectory). The main advantage of this bound is that it does not make any assumption on the range of the gradient sample. The variance of the sample can be upper bounded in the case of the REINFORCE [1] and the G(PO)MDP [3]/PGT [2] gradient estimators by using results from [21], already adapted for similar purposes in [15]. The G(PO)MDP/PGT estimator suffers from a smaller variance if compared with REINFORCE, and the variance bound is indeed tighter.

## 4.2 Hoeffding's Bound

By using Hoeffding's bound we obtain:

$$d_\delta = \mathbf{R}\sqrt{\frac{\log 2/\delta}{2}},$$

where $\mathbf{R}$ is the range of the gradient approximator, i.e., $|supp(\tilde{\nabla}_{\theta_i} J_\mu(\boldsymbol{\theta}))|$. For the class of policies we are considering, i.e., Gaussian with mean linear in the features, under some assumptions, the range can be upper bounded as follows:

**Lemma 4.3.** *For any Gaussian policy $\pi_\theta \sim \mathcal{N}(\boldsymbol{\theta}^T \boldsymbol{\phi}(s), \sigma^2)$, assuming that the action space is bounded ($\forall a \in \mathcal{A}, |a| \leq \overline{A}$) and the policy gradient is estimated on trajectories of length $H$, the range $\mathbf{R}$ of the policy gradient sample $\tilde{\nabla}_{\theta_i} J_\mu(\boldsymbol{\theta})$ can be upper bounded $\forall i = 1, \ldots, m$ and $\forall \boldsymbol{\theta}$ by*

$$\mathbf{R} \leq \frac{2H M_\phi \overline{A} R}{\sigma^2 (1 - \gamma)}.$$

As we will show in Section 5, a more practical solution (even if less rigorous) consists in computing the range as the difference between the largest and the smallest gradient sample seen during learning.

## 4.3 Empirical Bernstein's Bound

Tighter concentration bounds allow for smaller batch sizes (which result in more frequent policy updates) and larger step sizes, thus speeding up the learning process and improving long-time average performance. An empirical Bernstein bound from [22] allows to use sample variance instead of the variance bounds from [21] and to limit the impact of the gradient range. On the other hand, this bound does not satisfy Assumption 4.1, giving for the estimation error the following, more complex, expression:

$$\epsilon(N) = \frac{d_\delta}{\sqrt{N}} + \frac{f_\delta}{N},$$

where

$$d_\delta = \sqrt{2 S_N \ln 3/\delta}, \qquad\qquad f = 3\mathbf{R} \ln 3/\delta,$$

and $S_N$ is the sample variance of the gradient approximator. No reasonably simple closed-form solution is available in this case, requiring a linear search of the batch size $N^*$ maximizing $\Upsilon_\delta$. By adapting Assumption 3.7 to this case, a starting point for this search can be provided:

$$N \geq \left( \frac{d_\delta + \sqrt{d_\delta^2 + 4f_\delta \left\| \hat{\nabla}_{\boldsymbol{\theta}} J_\mu(\boldsymbol{\theta}) \right\|_\infty}}{2 \left\| \hat{\nabla}_{\boldsymbol{\theta}} J_\mu(\boldsymbol{\theta}) \right\|_\infty} \right)^2,$$

We also know that there is a unique maximum in $[N_0, +\infty)$ (see Appendix A for more details) and that $\Upsilon_\delta$ goes to 0 as N goes to infinity. Hence, to find the optimal batch size, it is enough to start from $N_0$ and stop as soon as the value of the cost function $\Upsilon(\Lambda^*, N)$ begins to decrease. Furthermore, the optimal step size is no longer constant: it can be computed with the expression given in Corollary 3.8 by setting $\epsilon := \epsilon(N^*)$. As for the Hoeffding's bound, the range $\mathbf{R}$ can be upper bounded exactly or estimated from samples.

Table 1: Improvement rate of the policy updates for different policy standard deviation $\sigma$, fixed batch size $N$ and fixed step size $\alpha$, using the G(PO)MDP gradient estimator.

|  |  | $\sigma = 0.5$ | | | $\sigma = 1$ | | |
|---|---|---|---|---|---|---|---|
|  |  | $N = 10000$ | $N = 1000$ | $N = 100$ | $N = 10000$ | $N = 1000$ | $N = 100$ |
|  | 1e-3 | 95.96% | 52.85% | 49.79% | 24.24% | 37.4% | 50.4% |
| $\alpha$ | 1e-4 | 100% | 73.27% | 51.41% | 100% | 27.03% | 46.08% |
|  | 1e-5 | 98.99% | 81.88% | 55.69% | 100% | 99.9% | 39.04% |
|  | 1e-6 | 100% | 83.88% | 58.44% | 100% | 100% | 86.04% |

Table 2: Average performance for different gradient estimators, statistical bounds and values of $\delta$. All results are averaged over 5 runs (95% confidence intervals are reported).

| Estimator | Bound | $\delta$ | $\overline{\Upsilon}$ | Confidence interval |
|---|---|---|---|---|
| REINFORCE | Chebyshev | 0.95 | -11.3266 | [-11.3277; -11.3256] |
| REINFORCE | Chebyshev | 0.75 | -11.4303 | [-11.4308; -11.4297] |
| REINFORCE | Chebyshev | 0.5 | -11.5947 | [-11.5958; -11.5937] |
| G(PO)MDP | Chebyshev | 0.95 | -10.6085 | [-10.6087; -10.6083] |
| G(PO)MDP | Chebyshev | 0.75 | -10.7141 | [-10.7145; -10.7136] |
| G(PO)MDP | Chebyshev | 0.5 | -10.9036 | [-10.904; -10.9031] |
| G(PO)MDP | Chebyshev | 0.25 | -11.2355 | [-11.2363; -11.2346] |
| G(PO)MDP | Chebyshev | 0.05 | -11.836 | [-11.8368; -11.8352] |
| G(PO)MDP | Hoeffding | 0.95 | -11.914 | [-11.9143; -11.9136] |
| G(PO)MDP | Bernstein | 0.95 | -10.2159 | [-10.2162; -10.2155] |
| G(PO)MDP | Hoeffding (empirical range) | 0.95 | -9.8582 | [-9.8589; -9.8574] |
| G(PO)MDP | Bernstein (empirical range) | 0.95 | -9.6623 | [-9.6619; -9.6627] |

## 5 Numerical Simulations

In this section, we test the proposed methods on the linear-quadratic Gaussian regulation (LQG) problem [23]. The LQG problem is defined by transition model $s_{t+1} \sim \mathcal{N}(s_t + a_t, \sigma_0^2)$, Gaussian policy $a_t \sim \mathcal{N}(\theta \cdot s, \sigma^2)$ and reward $r_t = -0.5(s_t^2 + a_t^2)$. In all our simulations we use $\sigma_0 = 0$, since all the noise can be modelled on the agent's side without loss of generality. Both action and state variables are bounded to the interval $[-2, 2]$ and the initial state is drawn uniformly at random. We use this task as a testing ground because it is simple, all the constants involved in our bounds can be computed exactly, and the true optimal parameter $\theta^*$ is available as a reference. We use a discount factor $\gamma = 0.9$, which gives an optimal parameter $\theta^* \approx -0.59$, corresponding to expected performance $J(\theta^*) \approx -13.21$. Coherently with the framework described in Section 1, we are interested both in the convergence speed and in the ratio of policy updates that does not result in a

worsening of the expected performance, which we will call *improvement ratio*. First of all, we want to analyze how the choice of fixed step sizes and batch sizes may affect the improvement ratio and how much it depends on the variability of the trajectories (that in this case is due to the variance of the policy). Table 1 shows the improvement ratio for two parameterizations ($\sigma = 0.5$ and $\sigma = 1$) when various constant step sizes and batch sizes are used, starting from $\theta = -0.55$ and stopping after a total of one million trajectories. As expected, small batch sizes combined with large step sizes lead to low improvement ratios. However, the effect is non-trivial and problem-dependent, justifying the need for an adaptive method.

We then proceed to test the methods described in Section 4. In the following simulations, we use $\sigma = 1$ and start from $\theta = 0$, stopping after a total of 30 million trajectories. Figure 1 shows the expected performance over sample trajectories for both the REINFORCE and G(PO)MDP gradient estimators, using Chebyshev's bound with different values of $\delta$. Expected performance is computed for each parameter update. Data are then scaled to account for the different batch sizes. In general, REINFORCE performs worse than G(PO)MDP due to its larger variance (in both cases the proper optimal baseline from [23] was used), and larger values of $\delta$ (the probability with which worsening updates are allowed to take place) lead to better performance. Notice that an improvement ratio of 1 is achieved also with large values of $\delta$. This is due to the fact that the bounds used in the development of our method are not tight. Being the method this conservative, in practical applications $\delta$ can be set to a high value to improve the convergence rate. Another common practice in empirical applications is to shrink confidence intervals through a scalar multiplicative factor. However, in this work we chose to not exploit this trick. Figure 2 compares the performance of the different concentration bounds described in the previous section, using always G(PO)MDP to estimate the gradient and $\delta = 0.95$. As expected, Bernstein's bound performs better than Chebyshev's, especially in the empirical range version. The rigorous version of Hoeffding's bound performs very poorly, while the one using the empirical range is almost as good as the corresponding Bernstein method. This is due to the fact that the bound on the gradient estimate range is very loose, since it accounts also for unrealistic combinations of state, action and reward. Finally, to better capture the performance of the different variants of the algorithm in a real-time scenario, we define a metric $\overline{\Upsilon}$, which is obtained by averaging the real performance (measured during learning) over all the trajectories, coherently with the cost function used to derive the optimal batch size. The results are reported in Table 2. In Appendix C we also show how the adaptive batch size evolves as the policy approaches the optimum.

## 6    Conclusions

We showed the relationship between the batch size and the step size in policy gradient approaches under Gaussian policies, and how their joint optimization can lead to parameters updates that guarantee with high probability a fixed improvement in the policy performance. In addition to the formal analysis, we proposed practical methods to compute the information required by the algorithms. Finally, we have proposed a preliminary evaluation on a simple control task. Future work should focus on developing more practical methods. It would also be interesting to investigate the extension of the proposed methodology to other classes of policies.

**Acknowledgments**

This research was supported in part by French Ministry of Higher Education and Research, Nord-Pas-de-Calais Regional Council and French National Research Agency (ANR) under project ExTra-Learn (n.ANR-14-CE24-0010-01).

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
