[Supplementary Material]

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

# A Proofs

In this appendix, we provide proofs for all the new theoretical results, including some auxiliary lemmas.

**Lemma A.1.** *For any initial state distribution $\mu$ and any pair of stationary Gaussian policies $\pi_{\boldsymbol{\theta}} \sim \mathcal{N}(\boldsymbol{\theta}^T \boldsymbol{\phi}(s), \sigma^2)$ and $\pi_{\boldsymbol{\theta}'} \sim \mathcal{N}(\boldsymbol{\theta}'^T \boldsymbol{\phi}(s), \sigma^2)$, so that $\boldsymbol{\theta}' = \boldsymbol{\theta} + \Lambda \nabla_{\boldsymbol{\theta}} J_\mu(\boldsymbol{\theta})$, and for any state $s$ and action $a$*

$$\pi(a|s, \boldsymbol{\theta}') - \pi(a|s, \boldsymbol{\theta}) \geq \nabla_{\boldsymbol{\theta}} \pi(a|s, \boldsymbol{\theta})^T \Lambda \nabla_{\boldsymbol{\theta}} J_\mu(\boldsymbol{\theta}) - \frac{M_\phi^2 \|\Lambda \nabla_{\boldsymbol{\theta}} J_\mu(\boldsymbol{\theta})\|_1^2}{\sqrt{2\pi}\sigma^3}$$

*Proof.* By exploiting Taylor's expansion

$$\begin{aligned}
\pi(a|s, \boldsymbol{\theta}') &= \pi(a|s, \boldsymbol{\theta} + \Lambda \nabla_{\boldsymbol{\theta}} J_\mu(\boldsymbol{\theta})) \\
&= \pi(a|s, \boldsymbol{\theta}) + \nabla_{\boldsymbol{\theta}} \pi(a|s, \boldsymbol{\theta})^T \Delta \boldsymbol{\theta} + R_1(\Delta \boldsymbol{\theta}),
\end{aligned}$$

where $R_1(\Delta \boldsymbol{\theta})$ is the remainder of the series which can be bounded as follows by exploiting Lemma 4.1 from [15] and Assumption 3.1:

$$\begin{aligned}
R_1(\Delta \boldsymbol{\theta}) &= \sum_{i=1}^m \sum_{j=1}^m \left. \frac{\partial^2 \pi(a|s, \boldsymbol{\theta})}{\partial \theta_i \partial \theta_j} \right|_{\boldsymbol{\theta}+c\Delta\boldsymbol{\theta}} \frac{\Delta\theta_i \Delta\theta_j}{1 + I(i=j)} \qquad \text{for some } c \in (0,1) \\
&\geq -\sum_{i=1}^m \sum_{j=1}^m \frac{|\phi_i(s)\phi_j(s)|}{\sqrt{2\pi}\sigma^3} \frac{\Delta\theta_i \Delta\theta_j}{1 + I(i=j)} \\
&= -\frac{1}{\sqrt{2\pi}\sigma^3} \sum_{i=1}^m \sum_{j=1}^m \frac{\alpha_i |\phi_i(s) \nabla_{\theta_i} J_\mu(\boldsymbol{\theta}) \alpha_j| \phi_j(s) |\nabla_{\theta_j} J_\mu(\boldsymbol{\theta})}{1 + I(i=j)} \\
&= -\frac{(|\nabla_{\boldsymbol{\theta}} J_\mu(\boldsymbol{\theta})|^T \Lambda |\boldsymbol{\phi}(s)|)^2}{\sqrt{2\pi}\sigma^3} \\
&\geq -\frac{M_\phi^2 \|\Lambda \nabla_{\boldsymbol{\theta}} J_\mu(\boldsymbol{\theta})\|_1^2}{\sqrt{2\pi}\sigma^3}.
\end{aligned}$$

It is now enough to apply this bound to Taylor's expansion. $\qquad\square$

**Lemma A.2.** *For any initial state distribution $\mu$ and any pair of stationary Gaussian policies $\pi_{\boldsymbol{\theta}} \sim \mathcal{N}(\boldsymbol{\theta}^T \boldsymbol{\phi}(s), \sigma^2)$ and $\pi_{\boldsymbol{\theta}'} \sim \mathcal{N}(\boldsymbol{\theta}'^T \boldsymbol{\phi}(s), \sigma^2)$, so that $\boldsymbol{\theta}' = \boldsymbol{\theta} + \Lambda \nabla_{\boldsymbol{\theta}} J_\mu(\boldsymbol{\theta})$*

$$\|\pi_{\boldsymbol{\theta}'} - \pi_{\boldsymbol{\theta}}\|_\infty^2 \leq \frac{M_\phi^2 \|\Lambda \nabla_{\boldsymbol{\theta}} J_\mu(\boldsymbol{\theta})\|_1^2}{\sigma^2}$$

*Proof.* By exploiting Pinsker's inequality [24]

$$\begin{aligned}
\|\pi_{\boldsymbol{\theta}'} - \pi_{\boldsymbol{\theta}}\|_\infty^2 &= \sup_s \|\pi_{\boldsymbol{\theta}'} - \pi_{\boldsymbol{\theta}}\|_\infty^2 \\
&\geq \sup_s 2H(\pi_{\boldsymbol{\theta}'} \| \pi_{\boldsymbol{\theta}}) \\
&= \sup_s \frac{1}{\sigma^2} \sum_i (\nabla_{\theta_i} J_\mu(\boldsymbol{\theta}) \alpha_i \phi_i(s))^2 \\
&\geq \frac{M_\phi^2 \|\Lambda \nabla_{\boldsymbol{\theta}} J_\mu(\boldsymbol{\theta})\|_1^2}{\sigma^2},
\end{aligned}$$

where $H(P\|Q)$ is the Kullback-Liebler divergence. $\qquad\square$

**Lemma 3.2.** *For any initial state distribution $\mu$ and any pair of stationary Gaussian policies $\pi_{\boldsymbol{\theta}} \sim \mathcal{N}(\boldsymbol{\theta}^T \boldsymbol{\phi}(s), \sigma^2)$ and $\pi_{\boldsymbol{\theta}'} \sim \mathcal{N}(\boldsymbol{\theta}'^T \boldsymbol{\phi}(s), \sigma^2)$, so that $\boldsymbol{\theta}' = \boldsymbol{\theta} + \Lambda \nabla_{\boldsymbol{\theta}} J_\mu(\boldsymbol{\theta})$, and under*

Assumption 3.1, the difference between the performance of $\pi_{\boldsymbol{\theta}'}$ and the one of $\pi_{\boldsymbol{\theta}}$ can be bounded below as follows:

$$J_{\mu}(\boldsymbol{\theta}') - J_{\mu}(\boldsymbol{\theta}) \geq \nabla_{\boldsymbol{\theta}} J_{\mu}(\boldsymbol{\theta})^{\mathsf{T}} \Lambda \nabla_{\boldsymbol{\theta}} J_{\mu}(\boldsymbol{\theta}) - \frac{\|\Lambda \nabla_{\boldsymbol{\theta}} J_{\mu}(\boldsymbol{\theta})\|_1^2 M_{\phi}^2}{(1-\gamma)\sigma^2}\left(\frac{1}{\sqrt{2\pi}\sigma}\int_{\mathcal{S}} d_{\mu}^{\pi_{\boldsymbol{\theta}}}(s)\int_{\mathcal{A}} Q^{\pi_{\boldsymbol{\theta}}}(s,a)\mathrm{d}a\mathrm{d}s + \frac{\gamma\|Q^{\pi_{\boldsymbol{\theta}}}\|_{\infty}}{2(1-\gamma)}\right),$$

where $\|Q^{\pi_{\boldsymbol{\theta}}}\|_{\infty}$ is the supremum norm of the Q-function: $\|Q^{\pi_{\boldsymbol{\theta}}}\|_{\infty} = \sup\limits_{s\in\mathcal{S}, a\in\mathcal{A}} Q^{\pi_{\boldsymbol{\theta}}}(s,a)$.

*Proof.* We plug the results of Lemmas A.1 and A.2 into Lemma 3.1 from [15]:

$$J_{\mu}(\boldsymbol{\theta}') - J_{\mu}(\boldsymbol{\theta}) \geq \frac{1}{1-\gamma}\int_{\mathcal{S}} d_{\mu}^{\pi_{\boldsymbol{\theta}}}(s)\int_{\mathcal{A}}(\pi(a|s,\boldsymbol{\theta}') - \pi(a|s,\boldsymbol{\theta}))Q^{\pi_{\boldsymbol{\theta}}}(s,a)\mathrm{d}a\mathrm{d}s$$

$$- \frac{\gamma}{2(1-\gamma)^2}\|\pi_{\boldsymbol{\theta}'} - \pi_{\boldsymbol{\theta}}\|_{\infty}^2\|Q^{\pi_{\boldsymbol{\theta}}}\|_{\infty}$$

$$\geq \frac{1}{1-\gamma}\int_{\mathcal{S}} d_{\mu}^{\pi_{\boldsymbol{\theta}}}(s)\int_{\mathcal{A}} \nabla_{\boldsymbol{\theta}}\pi(a|s,\boldsymbol{\theta})^T\Lambda\nabla_{\boldsymbol{\theta}} J_{\mu}(\boldsymbol{\theta})Q^{\pi_{\boldsymbol{\theta}}}(s,a)\mathrm{d}a\mathrm{d}s \qquad (1)$$

$$- \frac{M_{\phi}^2\|\Lambda\nabla_{\boldsymbol{\theta}} J_{\mu}(\boldsymbol{\theta})\|_1^2}{(1-\gamma)\sqrt{2\pi}\sigma^3}\int_{\mathcal{S}} d_{\mu}^{\pi_{\boldsymbol{\theta}}}(s)\int_{\mathcal{A}} Q^{\pi_{\boldsymbol{\theta}}}(s,a)\mathrm{d}a\mathrm{d}s \qquad (2)$$

$$- \frac{\gamma M_{\phi}^2\|\Lambda\nabla_{\boldsymbol{\theta}} J_{\mu}(\boldsymbol{\theta})\|_1^2}{2(1-\gamma)^2\sigma^2}\|Q^{\pi_{\boldsymbol{\theta}}}\|_{\infty} \qquad (3)$$

Term (1) can be simplified by using the Policy Gradient Theorem [2]:

$$\frac{1}{1-\gamma}\int_{\mathcal{S}} d_{\mu}^{\pi_{\boldsymbol{\theta}}}(s)\int_{\mathcal{A}} \nabla_{\boldsymbol{\theta}}\pi(a|s,\boldsymbol{\theta})^T\Lambda\nabla_{\boldsymbol{\theta}} J_{\mu}(\boldsymbol{\theta})Q^{\pi_{\boldsymbol{\theta}}}(s,a)\mathrm{d}a\mathrm{d}s$$

$$= \frac{\nabla_{\boldsymbol{\theta}} J_{\mu}(\boldsymbol{\theta})^T\Lambda}{1-\gamma}\int_{\mathcal{S}} d_{\mu}^{\pi_{\boldsymbol{\theta}}}(s)\int_{\mathcal{A}} \nabla_{\boldsymbol{\theta}}\pi(a|s,\boldsymbol{\theta})Q^{\pi_{\boldsymbol{\theta}}}(s,a)\mathrm{d}a\mathrm{d}s$$

$$= \nabla_{\boldsymbol{\theta}} J_{\mu}(\boldsymbol{\theta})^T\Lambda\nabla_{\boldsymbol{\theta}} J_{\mu}(\boldsymbol{\theta})$$

The proof now follows simply by rearranging terms. $\qquad\square$

**Theorem 3.3.** *For any initial state distribution $\mu$ and any pair of stationary Gaussian policies $\pi_{\boldsymbol{\theta}} \sim \mathcal{N}(\boldsymbol{\theta}^T\phi(s),\sigma^2)$ and $\pi_{\boldsymbol{\theta}'} \sim \mathcal{N}(\boldsymbol{\theta}'^T\phi(s),\sigma^2)$, so that $\boldsymbol{\theta}' = \boldsymbol{\theta} + \Lambda\nabla_{\boldsymbol{\theta}} J_{\mu}(\boldsymbol{\theta})$, and under Assumption 3.1, the difference between the performance of $\pi_{\boldsymbol{\theta}'}$ and the one of $\pi_{\boldsymbol{\theta}}$ can be bounded below as follows:*

$$J_{\mu}(\boldsymbol{\theta}') - J_{\mu}(\boldsymbol{\theta}) \geq \nabla_{\boldsymbol{\theta}} J_{\mu}(\boldsymbol{\theta})^{\mathsf{T}}\Lambda\nabla_{\boldsymbol{\theta}} J_{\mu}(\boldsymbol{\theta}) - c\|\Lambda\nabla_{\boldsymbol{\theta}} J_{\mu}(\boldsymbol{\theta})\|_1^2,$$

*where $c = \frac{RM_{\phi}^2}{(1-\gamma)^2\sigma^2}\left(\frac{|\mathcal{A}|}{\sqrt{2\pi}\sigma} + \frac{\gamma}{2(1-\gamma)}\right)$ and $|\mathcal{A}|$ is the volume of the action space.*

*Proof.* For every state $s\in\mathcal{S}$ and every action $a\in\mathcal{A}$, the Q-function belongs to $\left[-\frac{R}{1-\gamma}, \frac{R}{1-\gamma}\right]$. As a consequence, $\int_{\mathcal{A}} Q^{\pi_{\boldsymbol{\theta}}}(s,a)\mathrm{d}a \leq \frac{|\mathcal{A}|R}{1-\gamma}$ and $\|Q^{\pi_{\boldsymbol{\theta}}}\|_{\infty} \leq \frac{R}{1-\gamma}$. The proof follows from applying these bounds to the expression of Lemma 3.2. $\qquad\square$

**Corollary 3.4.** *The lower bound of Theorem 3.3 is maximized by the following non-scalar step size:*

$$\alpha_k^* = \begin{cases} \frac{1}{2c} & \text{if } k = \min\{\arg\max_i |\nabla_{\theta_i} J_{\mu}(\boldsymbol{\theta})|\}, \\ 0 & \text{otherwise}, \end{cases}$$

*which guarantees the following performance improvement: $J_{\mu}(\boldsymbol{\theta}') - J_{\mu}(\boldsymbol{\theta}) \geq \frac{\|\nabla_{\boldsymbol{\theta}} J_{\mu}(\boldsymbol{\theta})\|_{\infty}^2}{4c}$.*

*Proof.* We study the bound from Theorem 3.3, which we call $B$ for the duration of this proof, as a function of $\alpha_1,\ldots,\alpha_m$ constrained by $\alpha_i \geq 0\,\forall i = 1,\ldots,m$. The derivative of the bound w.r.t. $\alpha_k$ is:

$$\frac{\partial B}{\partial \alpha_k} = \nabla_{\theta_k} J_{\mu}(\boldsymbol{\theta})^2 - 2c|\nabla_{\theta_k} J_{\mu}(\boldsymbol{\theta})|\sum_{i=1}^{m}\left(\alpha_i|\nabla_{\theta_i} J_{\mu}(\boldsymbol{\theta})|\right).$$

Hence setting the gradient of $B$ to zero corresponds to the following problem:

$$\sum_{i=1}^{m} (\alpha_i |\nabla_{\theta_i} J_\mu(\boldsymbol{\theta})|) = \frac{|\nabla_{\theta_k} J_\mu(\boldsymbol{\theta})|}{2c} \qquad \forall k = 1, \ldots, m,$$

which is impossible (except from the case in which all the components of the gradient are equal in absolute value, which we address later). This means the function has no stationary points. In the most general case, the best we can do is to set to zero a single component of the gradient, i.e., partial maximization is possible only along a single dimension at a time. Candidates for the optimum are thus attained only on the boundary given by the constraint $\alpha_i \geq 0 \, \forall i = 1, \ldots, m$, and are obtained by setting to 0 all the components of the step size but one, say $\alpha_k$. Under this hypothesis the bound becomes:

$$B = \alpha_k \nabla_{\theta_k} J_\mu(\boldsymbol{\theta})^2 - c\alpha_k^2 \nabla_{\theta_k} J_\mu(\boldsymbol{\theta}),$$

which is maximized by setting $\alpha_k = \frac{1}{2c}$. This assignment yields:

$$J_\mu(\boldsymbol{\theta}') - J_\mu(\boldsymbol{\theta}) \geq \frac{\nabla_{\theta_k} J_\mu(\boldsymbol{\theta})^2}{4c}.$$

To obtain the global maximum is enough to select $k$ as to maximize the above quantity, which results in the following:

$$\alpha_k^* = \begin{cases} \frac{1}{2c} & \text{if } k = \arg\max_i |\nabla_{\theta_i} J_\mu(\boldsymbol{\theta})|, \\ 0 & \text{otherwise.} \end{cases}$$

In case the $\arg\max$ function returns more than one candidate, we have a looser condition. Suppose $|\hat{\nabla}_{\theta_i} J_\mu(\boldsymbol{\theta})| = |\hat{\nabla}_{\theta_j} J_\mu(\boldsymbol{\theta})| = \|\nabla_{\boldsymbol{\theta}} J_\mu(\boldsymbol{\theta})\|_\infty \quad \forall i, j \in I' \subseteq \{1, \ldots, m\}$. The bound becomes:

$$B = \|\nabla_{\boldsymbol{\theta}} J_\mu(\boldsymbol{\theta})\|_\infty \left[ \sum_{i \in I'} \alpha_i - c \left( \sum_{i \in I'} \alpha_i \right)^2 \right].$$

In this case the maximum is attained by any assignment $\Lambda$ such that:

$$\begin{cases} \sum_{i \in I'} \alpha_i = \frac{1}{2c} \\ \alpha_j = 0 & \forall j \notin I'. \end{cases}$$

A possible solution is to pick just one $k \in I'$, set $\alpha_k = 1/2c$ and all the other components of $\Lambda$ to 0. We adopt this solution in order to maintain the coordinate descent nature of the resulting algorithm, which is crucial for guaranteeing convergence. The non-zero component can be selected in any way, for instance by lexicographical order of the index. This gives us a more general expression for $\Lambda^*$:

$$\alpha_k^* = \begin{cases} \frac{1}{2c} & \text{if } k = \min\{\arg\max_i |\nabla_{\theta_i} J_\mu(\boldsymbol{\theta})|\}, \\ 0 & \text{otherwise.} \end{cases}$$

By substituting $\Lambda^*$ back into the bound, we obtain:

$$J_\mu(\boldsymbol{\theta}') - J_\mu(\boldsymbol{\theta}) \geq \frac{\|\nabla_{\boldsymbol{\theta}} J_\mu(\boldsymbol{\theta})\|_\infty^2}{4c}.$$

$\square$

**Theorem 3.6.** *Under the same assumptions of Theorem 3.3, and provided that a policy gradient estimate $\hat{\nabla}_{\boldsymbol{\theta}} J_\mu(\boldsymbol{\theta})$ is available, so that $\mathbb{P}(|\nabla_{\theta_i} J_\mu(\boldsymbol{\theta}) - \hat{\nabla}_{\theta_i} J_\mu(\boldsymbol{\theta})| \geq \epsilon_i) \leq \delta, \, \forall i = 1, \ldots, m$, the difference between the performance of $\pi_{\boldsymbol{\theta}'}$ and the one of $\pi_{\boldsymbol{\theta}}$ can be bounded below with probability at least $(1 - \delta)^m$ as follows:*

$$J_\mu(\boldsymbol{\theta}') - J_\mu(\boldsymbol{\theta}) \geq \underline{\hat{\nabla}_{\boldsymbol{\theta}} J_\mu(\boldsymbol{\theta})}^T \Lambda \underline{\hat{\nabla}_{\boldsymbol{\theta}} J_\mu(\boldsymbol{\theta})} - c \left\| \Lambda \overline{\hat{\nabla}_{\boldsymbol{\theta}} J_\mu}(\boldsymbol{\theta}) \right\|_1^2,$$

*where $c$ is defined as in Theorem 3.3.*

*Proof.* The proof immediately follows from Theorem 3.3 and the definition of $\hat{\nabla}_{\boldsymbol{\theta}} J_\mu(\boldsymbol{\theta})$ and $\overline{\hat{\nabla}_{\boldsymbol{\theta}} J_\mu}(\boldsymbol{\theta})$. Note that the saturation to 0 in $\hat{\nabla}_{\boldsymbol{\theta}} J_\mu(\boldsymbol{\theta})$ is necessary since $\nabla_{\boldsymbol{\theta}} J_\mu(\boldsymbol{\theta})$ is taken with absolute value in the negative term of the original bound. $\square$

**Corollary 3.8.** *The performance lower bound of Theorem 3.6 is maximized under Assumption 3.7 by the following non-scalar step size:*

$$\alpha_k^* = \begin{cases} \frac{\left(\left\|\hat{\nabla}_{\boldsymbol{\theta}} J_\mu(\boldsymbol{\theta})\right\|_\infty - \epsilon\right)^2}{2c\left(\left\|\hat{\nabla}_{\boldsymbol{\theta}} J_\mu(\boldsymbol{\theta})\right\|_\infty + \epsilon\right)^2} & \text{if } k = \min\left\{\arg\max_i |\hat{\nabla}_{\theta_i} J_\mu(\boldsymbol{\theta})|\right\}, \\ 0 & \text{otherwise,} \end{cases}$$

*which guarantees with probability $(1-\delta)^m$ a performance improvement*

$$J_\mu(\boldsymbol{\theta}') - J_\mu(\boldsymbol{\theta}) \geq \frac{\left(\left\|\hat{\nabla}_{\boldsymbol{\theta}} J_\mu(\boldsymbol{\theta})\right\|_\infty - \epsilon\right)^4}{4c\left(\left\|\hat{\nabla}_{\boldsymbol{\theta}} J_\mu(\boldsymbol{\theta})\right\|_\infty + \epsilon\right)^2}.$$

*Proof.* The derivation of the optimal step size is similar to the one of Corollary 3.4. The only difference is that we must select the parameter to update as follows:

$$\theta_k \mid k = \min\left\{\arg\max_i \left\{\frac{\max(|\hat{\nabla}_{\theta_i} J_\mu(\boldsymbol{\theta})| - \epsilon_i, 0)^2}{4c(|\hat{\nabla}_{\theta_i} J_\mu(\boldsymbol{\theta})| + \epsilon_i)^2}\right\}\right\}.$$

In the case $\epsilon_1 = \epsilon_2 = \ldots = \epsilon_m \triangleq \epsilon$, which can always be obtained by setting $\epsilon = \max_i \epsilon_i$, the criterion can be simplified as:

$$k = \min\left\{\arg\max_i \left\{\frac{\max(|\hat{\nabla}_{\theta_i} J_\mu(\boldsymbol{\theta})| - \epsilon, 0)^2}{4c(|\hat{\nabla}_{\theta_i} J_\mu(\boldsymbol{\theta})| + \epsilon)^2}\right\}\right\}.$$

Then, since we are already maximizing, the $\max(\cdot, 0)$ operator can be removed (under Assumption 4.2):

$$k = \min\left\{\arg\max_i \left\{\frac{(|\hat{\nabla}_{\theta_i} J_\mu(\boldsymbol{\theta})| - \epsilon)^2}{4c(|\hat{\nabla}_{\theta_i} J_\mu(\boldsymbol{\theta})| + \epsilon)^2}\right\}\right\}.$$

Being the objective function monotonic non-decreasing in $|\hat{\nabla}_{\theta_k} J_\mu(\boldsymbol{\theta})|$, we can simply select $k$ as to maximize $|\hat{\nabla}_{\theta_k} J_\mu(\boldsymbol{\theta})|$, obtaining:

$$\alpha_k^* = \begin{cases} \frac{\left(\left\|\hat{\nabla}_{\boldsymbol{\theta}} J_\mu(\boldsymbol{\theta})\right\|_\infty - \epsilon\right)^2}{2c\left(\left\|\hat{\nabla}_{\boldsymbol{\theta}} J_\mu(\boldsymbol{\theta})\right\|_\infty + \epsilon\right)^2} & \text{if } k = \min\left\{\arg\max_i |\hat{\nabla}_{\theta_i} J_\mu(\boldsymbol{\theta})|\right\}, \\ 0 & \text{otherwise.} \end{cases}$$

By substituting $\Lambda^*$ back into the bound, we obtain:

$$J_\mu(\boldsymbol{\theta}') - J_\mu(\boldsymbol{\theta}) \geq \frac{\left(\left\|\hat{\nabla}_{\boldsymbol{\theta}} J_\mu(\boldsymbol{\theta})\right\|_\infty - \epsilon\right)^4}{4c\left(\left\|\hat{\nabla}_{\boldsymbol{\theta}} J_\mu(\boldsymbol{\theta})\right\|_\infty + \epsilon\right)^2}.$$

$\square$

**Theorem 4.2.** *Under the hypotheses of Theorem 3.3 and Assumption 4.1, the cost-sensitive performance improvement measure $\Upsilon_\delta$, as defined in Definition 4.1, is maximized by the following step size and batch size:*

$$\alpha_k^* = \begin{cases} \frac{(13 - 3\sqrt{17})}{4c} & \text{if } k = \min\left\{\arg\max_i |\hat{\nabla}_{\theta_i} J_\mu(\boldsymbol{\theta})|\right\}, \\ 0 & \text{otherwise,} \end{cases} \qquad N^* = \left\lceil \frac{(13 + 3\sqrt{17})d_\delta^2}{2\left\|\hat{\nabla}_{\boldsymbol{\theta}} J_\mu(\boldsymbol{\theta})\right\|_\infty^2} \right\rceil,$$

*where $c = \frac{RM_\phi^2}{(1-\gamma)^2\sigma^2}\left(\frac{|\mathcal{A}|}{\sqrt{2\pi}\sigma} + \frac{\gamma}{2(1-\gamma)}\right)$. This choice guarantees with probability $(1-\delta)^m$ a performance improvement of:*

$$J_\mu(\boldsymbol{\theta}') - J_\mu(\boldsymbol{\theta}) \geq \frac{393 - 95\sqrt{17}}{8}\left\|\hat{\nabla}_{\boldsymbol{\theta}} J_\mu(\boldsymbol{\theta})\right\|_\infty^2 \geq 0.16\left\|\hat{\nabla}_{\boldsymbol{\theta}} J_\mu(\boldsymbol{\theta})\right\|_\infty^2.$$

*Proof.* We first optimize the cost function $\Upsilon_\delta$ w.r.t $\Lambda$. Since $\Upsilon_\delta$ is just the bound from Theorem 3.6 divided by $N$, we can use the result from Corollary 3.8, which under Assumption 4.1 can be expressed as:

$$\alpha_k^* = \begin{cases} \frac{\left(\left\|\hat{\nabla}_{\boldsymbol{\theta}} J_\mu(\boldsymbol{\theta})\right\|_\infty - d_\delta/\sqrt{N}\right)^2}{2c\left(\left\|\hat{\nabla}_{\boldsymbol{\theta}} J_\mu(\boldsymbol{\theta})\right\|_\infty + d_\delta/\sqrt{N}\right)^2} & \text{if } k = \min\left\{\arg\max_i |\hat{\nabla}_{\theta_i} J_\mu(\boldsymbol{\theta})|\right\}, \\ 0 & \text{otherwise,} \end{cases}$$

which yields:

$$\Upsilon_\delta(\Lambda^*, N) = \frac{\left(\left\|\hat{\nabla}_{\boldsymbol{\theta}} J_\mu(\boldsymbol{\theta})\right\|_\infty - d_\delta/\sqrt{N}\right)^4}{4c\left(\left\|\hat{\nabla}_{\boldsymbol{\theta}} J_\mu(\boldsymbol{\theta})\right\|_\infty + d_\delta/\sqrt{N}\right)^2 N}.$$

To justify the use of Corollary 3.8, our $N^*$ must be compliant with 3.7, which, under Assumption 4.1, translates into the following constraint:

$$N \geq N_0 := \frac{d_\delta^2}{\left\|\hat{\nabla}_{\boldsymbol{\theta}} J_\mu(\boldsymbol{\theta})\right\|_\infty^2}.$$

By computing the derivative $\partial \Upsilon_\delta / \partial N$ we find just two stationary points in $[N_0, +\infty)$: the first one is $N_0$ itself, which is a minimum ($\Upsilon_\delta(\Lambda^*, N_0) = 0$ and $\Upsilon_\delta$ is non-negative); the other one is our optimal batch size:

$$N^* = \frac{(13 + 3\sqrt{17})d_\delta^2}{2\left\|\hat{\nabla}_{\boldsymbol{\theta}} J_\mu(\boldsymbol{\theta})\right\|_\infty^2}.$$

Since $\Upsilon_\delta(\Lambda^*, N_1) = 0$, $\lim_{N \to +\infty} \Upsilon_\delta(\Lambda^*, N) = 0$, and $\Upsilon_\delta$ is continuous and differentiable in $[N_0, +\infty)$, $N^*$ is indeed the global maximum in the region of interest. We can now substitute $N^*$ into $\Lambda^*$ to obtain:

$$\alpha_k^* = \begin{cases} \frac{(13 - 3\sqrt{17})}{4c} & \text{if } k = \min\left\{\arg\max_i |\hat{\nabla}_{\theta_i} J_\mu(\boldsymbol{\theta})|\right\}, \\ 0 & \text{otherwise,} \end{cases}$$

and into $\Upsilon_\delta(\Lambda^*, N^*)$ to obtain:

$$\Upsilon^* = \frac{(4977 - 1207\sqrt{17})\left\|\hat{\nabla}_{\boldsymbol{\theta}} J_\mu(\boldsymbol{\theta})\right\|_\infty^4}{32d_\delta^2}.$$

Finally

$$J_\mu(\boldsymbol{\theta}') - J_\mu(\boldsymbol{\theta}) \geq N^*\Upsilon^* = \frac{393 - 95\sqrt{17}}{8}\left\|\hat{\nabla}_{\boldsymbol{\theta}} J_\mu(\boldsymbol{\theta})\right\|_\infty^2$$

with probability at least $(1 - \delta)^m$.

$\square$

**Lemma 4.3.** *For any Gaussian policy $\pi_\theta \sim \mathcal{N}(\boldsymbol{\theta}^T\boldsymbol{\phi}(s), \sigma^2)$, assuming that the action space is bounded ($\forall a \in \mathcal{A}, |a| \leq \overline{A}$) and the policy gradient is estimated on trajectories of length $H$, the range $\mathbf{R}$ of the policy gradient sample $\hat{\nabla}_{\theta_i} J_\mu(\boldsymbol{\theta})$ can be upper bounded $\forall i = 1, \ldots, m$ and $\forall \boldsymbol{\theta}$ by*

$$\mathbf{R} \leq \frac{2HM_\phi \overline{A} R}{\sigma^2(1 - \gamma)}.$$

*Proof.* We focus on the REINFORCE[1] gradient estimator:

$$\hat{\nabla}_{\boldsymbol{\theta}} J_\mu^{RF}(\boldsymbol{\theta}) = \frac{1}{N}\sum_{n=1}^N\left(\sum_{k=1}^H \nabla_{\boldsymbol{\theta}} \log \pi(a_k^n|s_k^n, \boldsymbol{\theta})\left(\sum_{l=1}^H \gamma^{l-1}r_l^n - b\right)\right),$$

and the G(PO)MDP[3]/PGT[2] gradient estimator:

$$\hat{\nabla}_{\boldsymbol{\theta}} J_{\mu}^{PGT}(\boldsymbol{\theta}) = \hat{\nabla}_{\boldsymbol{\theta}} J_{\mu}^{G(PO)MDP}(\boldsymbol{\theta}) = \frac{1}{N} \sum_{n=1}^{N} \left( \sum_{k=1}^{H} \nabla_{\boldsymbol{\theta}} \log \pi(a_k^n | s_k^n, \boldsymbol{\theta}) \left( \sum_{l=k}^{H} \gamma^{l-1} r_l^n - b_l^n \right) \right).$$

In both cases, the i-th component of the single sample can be bounded in absolute value as

$$\sum_{k=1}^{H} |\nabla_{\theta_i} \log \pi(a_k^n | s_k^n, \boldsymbol{\theta})| \frac{R}{1-\gamma}.$$

In the case of Gaussian policy, bounded action space ($|a| \leq \overline{A}$) and under Assumption 3.1, in the most general case, the range of the term $\nabla_{\theta_i} \log \pi(a_k^n | s_k^n, \boldsymbol{\theta})$ can be bounded as $\frac{2M_\phi \overline{A}}{\sigma^2}$. Finally

$$\mathbf{R} \leq \frac{2HM_\phi \overline{A} R}{\sigma^2 (1-\gamma)}.$$

$\square$

# B  More on Statistical Bounds

**Chebyshev's bound.**   Using results from [21] as adapted in [15], we can give explicit formulations for the optimal batch size. When the REINFORCE [1] gradient estimator ($\hat{\nabla}_{\boldsymbol{\theta}} J_{\mu}^{RF}(\boldsymbol{\theta})$) is used to estimate the gradient, we can use Lemma 5.4 from [15] to bound the estimation error as:

$$\epsilon \leq \frac{1}{\sqrt{N}} \left( \frac{RM_\phi (1-\gamma^H)}{\sigma(1-\gamma)} \sqrt{\frac{H}{\delta}} \right),$$

which gives an optimal batch size:

$$N^* = \frac{(13 + 3\sqrt{17}) R^2 M_\phi^2 H (1-\gamma^H)^2}{2\delta\sigma^2 (1-\gamma)^2 \left\| \hat{\nabla}_{\boldsymbol{\theta}} J_{\mu}^{RF}(\boldsymbol{\theta}) \right\|_{\infty}^2}.$$

Similarly, when the G(PO)MDP/PGT gradient estimator ($\hat{\nabla}_{\boldsymbol{\theta}} J_{\mu}^{PGT}(\boldsymbol{\theta})$) is used, using Lemma 5.5 from [15] we have:

$$\epsilon \leq \frac{1}{\sqrt{N}} \left( \frac{RM_\phi}{\sigma(1-\gamma)} \sqrt{\frac{1}{\delta} \left[ \frac{1-\gamma^{2H}}{1-\gamma^2} + H\gamma^{2H} - 2\gamma^H \frac{1-\gamma^H}{1-\gamma} \right]} \right)$$

and

$$N^* = \frac{(13 + 3\sqrt{17}) R^2 M_\phi^2 \left[ \frac{1-\gamma^{2H}}{1-\gamma^2} + H\gamma^{2H} - 2\gamma^H \frac{1-\gamma^H}{1-\gamma} \right]}{2\delta\sigma^2 (1-\gamma)^2 \left\| \hat{\nabla}_{\boldsymbol{\theta}} J_{\mu}^{PGT}(\boldsymbol{\theta}) \right\|_{\infty}^2}.$$

These results are independent on the baseline used in the gradient estimation.

**Hoeffding's bound.**   The explicit expression for the optimal batch size using Hoeffding's bound is:

$$N^* = \frac{(13 + 3\sqrt{17}) \mathbf{R}^2 \log 2/\delta}{4 \left\| \hat{\nabla}_{\boldsymbol{\theta}} J_{\mu}(\boldsymbol{\theta}) \right\|_{\infty}^2}.$$

**Empirical Bernstein's bound.** Using the empirical Bernstein's bound from [22], the cost function to optimize becomes (already optimized w.r.t the step size):

$$\Upsilon_\delta(\Lambda^*, N) = \frac{\left(\left\|\hat{\nabla}_{\boldsymbol{\theta}} J_\mu(\boldsymbol{\theta})\right\|_\infty - \sqrt{\frac{2S_N \ln{^3/\delta}}{N}} - \frac{3\mathbf{R}\ln{^3/\delta}}{N}\right)^4}{4cN\left(\left\|\hat{\nabla}_{\boldsymbol{\theta}} J_\mu(\boldsymbol{\theta})\right\|_\infty + \sqrt{\frac{2S_N \ln{^3/\delta}}{N}} + \frac{3\mathbf{R}\ln{^3/\delta}}{N}\right)^2}.$$

First of all, Assumption 3.7 gives the following constraint:

$$N \geq N_0 := \left(\frac{d_\delta + \sqrt{d_\delta^2 + 4f_\delta\left\|\hat{\nabla}_{\boldsymbol{\theta}} J_\mu(\boldsymbol{\theta})\right\|_\infty}}{2\left\|\hat{\nabla}_{\boldsymbol{\theta}} J_\mu(\boldsymbol{\theta})\right\|_\infty}\right)^2.$$

By computing the derivative w.r.t. $N$ we obtain:

$$-\left(d_\delta\sqrt{N} + f_\delta - \left\|\hat{\nabla}_{\boldsymbol{\theta}} J_\mu(\boldsymbol{\theta})\right\|_\infty N\right)^3 \times$$

$$\times \frac{\left(2d_\delta^2 N + 5d_\delta f_\delta \sqrt{N} + 3d_\delta\left\|\hat{\nabla}_{\boldsymbol{\theta}} J_\mu(\boldsymbol{\theta})\right\|_\infty N^{3/2} + 3f_\delta^2 + 6f\left\|\hat{\nabla}_{\boldsymbol{\theta}} J_\mu(\boldsymbol{\theta})\right\|_\infty N - \left\|\hat{\nabla}_{\boldsymbol{\theta}} J_\mu(\boldsymbol{\theta})\right\|_\infty^2 N^2\right)}{4cN^4\left(d\sqrt{N} + f + \left\|\hat{\nabla}_{\boldsymbol{\theta}} J_\mu(\boldsymbol{\theta})\right\|_\infty N\right)^3}.$$

The left side of the numerator gives again $N_0$. By applying Descartes' rule of signs to the left term, seen as a polynomial in $\sqrt{N}$, we see that it gives just one root. The exact expression of this maximum is too big to be reported, but its uniqueness justifies the methodology proposed for the search of $N^*$.

## C  More on Numerical Simulations

We show how the adaptive batch size evolves during the learning process. The following plots are relative to the experiments reported in Figure 2. Simulations are stopped after a total of 30 million trajectories, so different methods perform a different number of iterations. As the policy approaches an optimum, larger and larger batch sizes need to be employed to guarantee a constant improvement. We can see how less conservative methods, such as the ones using empirical ranges, show larger oscillations in the batch size.

Figure 3: Adaptive batch size over learning iterations, using Chebyshev's bound.

Figure 4: Adaptive batch size over learning iterations, using Hoeffding's bound.

Figure 5: Adaptive batch size over learning iterations, using Hoeffding's bound with empirical range.

Figure 6: Adaptive batch size over learning iterations, using empirical Bernstein's bound.

Figure 7: Adaptive batch size over learning iterations, using empirical Bernstein's bound with empirical range.