[Reviews · NeurIPS 2017]

Reviewer 1



Summary: This paper derives conditions for guaranteed improvement when using policy gradient methods. These conditions are for stochastic gradient estimates and also bound the amount of improvement with high probability. The authors then show how these bounds can be optimized by properly selecting the step size and batch size parameters. This is in contrast to previous work that only considers how the step size can be optimized. The result is an algorithm that can guarantee improvement with high probability. Note: I did not verify all of the math, but at an intuitive level the theorems and lemmas in the main body are believable. Major: 1. What am I supposed to take away from this paper? Am I supposed to think "I should use policy gradient methods with their safe step size & batch size approach if I need a safe RL algorithm"? Or, am I supposed to think "This paper is presenting some interesting theoretical insights into how step sizes and batch sizes impact policy gradient methods"? If the answer is the former, then the results are abysmally bad. The axes are all scaled by 10^7 trajectories. Even the fastest learning methods are therefore impractically slow. This doesn't appear to be a practical algorithm. Compare to the results in your reference [23], where high confidence improvement is achieved using hundreds to thousands of episodes. So, the proposed algorithm does not appear to be practical or competitive as a practical method. This makes me think that the contribution is the latter: a theoretical discussion about step sizes, batch sizes, and how they relate to guaranteed improvement. This is interesting and I would recommend acceptance if this was the pitch of the paper. However, this is not made clear in the introduction. Until I reached the empirical results I thought that the authors were proposing a practical algorithm. I propose that the introduction should be modified to make it clear what the contribution is: are you presenting theoretical results (e.g., like a PAC RL paper), or are you proposing a practical algorithm? If the answer is the latter, then you need to compare to existing works or explain why it is not an issue that your method requires so much data. 2. The assumption that the policy is Gaussian is crippling. In many of the real-world applications of RL (e.g., dialogue systems) the actions are discrete, and so the policy is not Gaussian. Does this extend to other policy forms, like softmax policies? Is this extension straightforward, or are there additional challenges? Medium: 1. Line 92. You say that convergence is assured because the angular difference is at most \pi/2. A similar claim was made by Peters and Schaal in their natural actor-critic paper, and it later turned out to be false (See "GeNGA: A generalization of natural gradient ascent with positive and negative convergence results" by Thomas at ICML 2014). Your claim is different from Peters and Schaal's, but due to the history of these off-hand claims tending to be false, you should provide a proof or a reference for this claim. 2. You use concentration bounds that tend to be wildly over-conservative (Chebyshev & Hoeffding) because they must hold for a broad class of distributions. This is obviously the correct first step. However, have you considered using bounds that are far tighter (but make false distributional assumptions), like Student's t-test? If the gradient estimate is the mean of n i.i.d gradient estimates, then by the Central Limit Theorem the gradient estimate is approximately normally distributed, and so Student's t-test is reasonable. Empirically, does it tend to result in improvements with high probability while requiring far less data? It seems like a practical implementation should not use Hoeffding / Chebyshev / Bernstein... 3. Page 7 is one log paragraph. Minor: 1. Line 74 "paramters" 2. Capitalization in section titles is inconsistent (e.g., section 3 capitalizes Policies but not Size or Scalar). 3. Line 91 "but, being the \alpha_i non-negative". Grammar error here. 4. Integrals look better with a slight space before the dx terms: Xf(x)dx. The $\,$ here helps split the terms visually. 5. Line 102 "bound requires to compute" - requires who or what to compute something? Grammar. 6. The line split on line 111 is awkward. 7. Equation below 113, use \text{} for "if" and "otherwise". 8. Below 131, $\widehat$ looks better on the $\nabla$ symbol, and use $\max$ for max. 9. Line 140 say what you mean by the "optimal step size" using an equation. 10. Line 192, don't use contractions in technical writing (i.e., don't write "doesn't", write "does not"). AFTER AUTHOR REBUTTAL: My overall review has not changed. I've increased to accept (7 from 6) since the authors are not claiming that they are presenting a practical algorithm, but rather presenting interesting theoretical results.

Reviewer 2



I enjoyed this paper, understood the importance of the basic idea, and think it is worthy of publication. Some comments: Some of the writing sounds strange, just in the introduction we have: "performance of such policies strictly depends on the model accuracy" strictly? maybe strongly is better. "think about equipment wear in smart manufacturing", I think a 'for example' would suffice. "hinder the performance loss" reduce the performance loss? There are some other cases of unusual wording in the paper which make it a little more difficult to read. Why is diag(\alpha_i) = \Lambda when a much more easily understood notation would be \diag(\lambda_i) = \Lambda? Throughout the paper argmax is used as a function, but argmax actually returns the *set* of all indices that achieve the max. Usually this would be an unimportant distinction but in this paper the return value from 'argmax' is set to a non-zero value and all other indices set to zero, with no consideration for what to do in cases with more than one element in the argmax. Theorem 3.3 is referenced several times but does not appear anywhere in the paper or supplementary material. The proof for Corollary 3.1 needs fleshing out "The function has no stationary points. Moreover, except from degenerate cases, partial maximization is possible only along a single dimension at a time. Candidates for the optimum are thus attained only on the boundary, and are obtained by setting to 0 all the components of the step size but one" is not sufficient since it requires too much of the reader (why does it have no stationary points? why is maximization possible only along a single dimension? etc.). Table 2: "5% confidence intervals are reported", surely this is meant to be 95% confidence intervals? If not then the results should be changed to be 95% intervals, 5% intervals are meaningless for all practical purposes. Sect 5: "We then proceed to test the methods described in Section 5." Should be section 4 I think. === review update === Happy to recommend for publication.

Reviewer 3



--Brief summary of the paper: The paper proposes a method to optimize the step size and batch size for policy gradient methods such that monotonic improvement of a policy is guaranteed with high probability. The paper focuses on a class of linear-in-parameter Gaussian policies and first derives an adaptive step-size which results in a coordinate descent like update. Then, the paper derives adaptive batch sizes for gradient estimation using different concentration bounds. Empirical behaviors of these adaptive batch sizes are evaluated and analyzed on a simple experiment. --Major comments and questions: The paper tackles important issues of policy gradient methods which are to choose step size and batch size, which are usually tackled by trial-and-error. The paper provides rigorous theoretical results for a class of linear Gaussian policy function. The writing is also easy to follow and understand. The paper's theoretical contribution to policy gradient framework is significant, although the practical contribution is mild. However, I have the following questions. 1) (Important question) In Theorem 4.1, the adaptive batch size N* depends on the gradient estimate. However, the gradient is usually estimated from a batch of trajectory samples. Is the gradient estimated before determining N*? If it is so, then what is the adaptive batch size N* used for? The same question is also asked for the initial search solution in Section 4.3 which again depends on the gradient estimate. 2) In the paper, the variance of Gaussian policy (sigma) is fixed and not learned. Intuitively, Gaussian policy with large variance often require a large batch size to accurately estimate the gradient. Does the theorem also support this? The paper should mention this viewpoint as well. 3) Since the performance improvement in Definition 4.1 is also a function of the policy variance through the constant $c$, can the theorems be extended to adaptively adjust the policy variance? This would be an important extension since in practice the policy variance should not be a common fixed value and should be determined through learning. --Minor suggestions: 1) Natural policy gradient and actor-critic are important subclasses of policy gradient methods. Is it straightforward to apply the method to them? (For actor-critic I suppose the theorems need to consider an error bound of the Q-function as well, and this might be a promising future direction.) 2) The state value function V(s) used in line 66 should be properly defined. 3) Having equation numbers will make the paper easier to read. 4) Some typos: Line 72, paramters -> parameters. Line 24: been -> be. Line 226: s_t ~ N(…) -> s_t+1 ~ N(…). Line 265: methods -> method. Duplicated references for [16] and [17], [21] and [22].